# SEDONA: Search for Decoupled Neural Networks toward Greedy Block-wise Learning

**Myeongjang Pyeon, Jihwan Moon, Taeyoung Hahn, and Gunhee Kim**
Seoul National University, Seoul, Korea
{mjpyeon,mnmjh1215,taeyounghahn,gunhee}@snu.ac.kr
https://vision.snu.ac.kr/projects/sedona

## Abstract

Backward locking and update locking are well-known sources of inefficiency in backpropagation that prevent from concurrently updating layers. Several works have recently suggested using local error signals to train network blocks asynchronously to overcome these limitations. However, they often require numerous iterations of trial-and-error to find the best configuration for local training, including how to decouple network blocks and which auxiliary networks to use for each block. In this work, we propose a differentiable search algorithm named SEDONA to automate this process. Experimental results show that our algorithm can consistently discover transferable decoupled architectures for VGG and ResNet variants, and significantly outperforms the ones trained with end-to-end backpropagation and other state-of-the-art greedy-leaning methods in CIFAR-10, Tiny-ImageNet and ImageNet.

## 1 Introduction

Backpropagation (Rumelhart et al., 1986) has made a significant contribution to the success of deep learning as the core learning algorithm for SGD-based optimization. However, backpropagation is sequential in nature and supports only synchronous weight updates. Specifically, the limited concurrency in backpropagation breaks down into two locking problems (Jaderberg et al., 2017). First, update locking – a forward pass must complete first before any weight update. Second, backward locking – gradient computation of upper layers must precede that of lower layers. Also, backpropagation may be biologically implausible since the human brain prefers local learning rules without the global movement of error signals (Crick (1989); Marblestone et al. (2016); Lillicrap et al. (2020)).

Greedy block-wise learning is a competitive alternative to backpropagation that overcomes these limitations. It splits layers into a stack of gradient-isolated blocks, each of which is trained with local error signals. Therefore, it is possible to simultaneously compute the gradients for different network components with more fine-grained locks. Limiting the depth of error propagation graphs also reduces the vanishing gradient and increases memory efficiency. Recently, Belilovsky et al. (2019), Nøkland & Eidnes (2019), Belilovsky et al. (2020), and Löwe et al. (2019) empirically demonstrated that greedy block-wise learning could yield competitive performance to end-to-end backpropagation.

However, greedy block-wise learning introduces a group of new architectural decisions. Let us consider a case where we want to decouple an $L$-layer network into $K$ blocks for a given $K \in \{1, \ldots, L\}$. Then, the number of all possible groupings is $\binom{L-1}{K-1}$. If we want to choose one of $M$ candidates of auxiliary networks to generate local error gradients, we would have to consider $\binom{L-1}{K-1} M^{K-1}$ different configurations. If local signals are not representative of the global goal, then the final performance would be damaged significantly.

In this work, we introduce a novel search method named SEDONA (SEarching for DecOupled Neural Architectures), which allows efficient search of decoupled neural architectures toward greedy block-wise learning. Given a base neural network, SEDONA optimizes the validation loss by grouping layers into blocks and selecting the best auxiliary network for each block. Inspired by DARTS (Liu et al., 2019), we first relax the decision variables representing error propagation graphs and

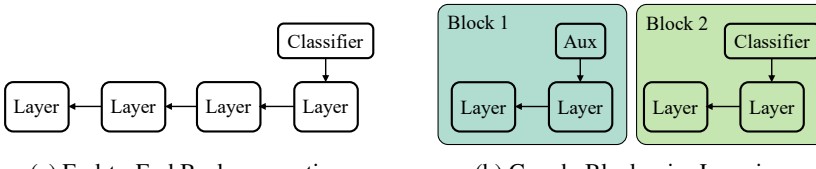

(a) End-to-End Backpropagation        (b) Greedy Block-wise Learning

Figure 1: Conceptual comparison of the backward computation graph between (a) end-to-end back-propagation and (b) greedy block-wise learning with $K = 2$.

auxiliary networks to continuous domains. We then formulate a bilevel optimization problem for the decision variables, which is solved via gradient descent.

Our key contributions are summarized as follows.

1. To the best of our knowledge, this work is the first attempt to automate the discovery of decoupling neural networks for greedy block-wise learning. We propose an efficient search method named SEDONA, which finds decoupled error propagation graphs and auxiliary heads suitable for successful greedy training.

2. Through extensive experiments on image classification tasks, we show that our *locally optimized* networks outperform not only the ones trained with end-to-end backpropagation but also two state-of-the-art greedy-learning methods of DGL (Belilovsky et al., 2020) and PredSim (Nøkland & Eidnes, 2019) in CIFAR-10, Tiny-ImageNet and ImageNet.

3. SEDONA discovers decoupled architectures for VGG (Simonyan & Zisserman, 2015) and ResNet (He et al., 2016) variants with only $0.25\times$ width on CIFAR-10. The discovered networks are transferable to Tiny-ImageNet and ImageNet in which they are high-performing enough to beat backpropagation and other greedy-leaning methods. It means that no search is required for every pair of a network and a dataset.

4. Finally, based on experimental results, we analyze the common characteristics among the architectures that are favorable to greedy-learning, such as avoiding the shallow first block and using deeper auxiliary networks for lower blocks.

## 2   PROBLEM STATEMENT AND MOTIVATION

In typical neural network training, backpropagation computes the gradients of weights with respect to the global loss by the chain rule (Rumelhart et al., 1986). On the other hand, in greedy block-wise learning (Löwe et al., 2019; Belilovsky et al., 2020), the network is split into several subnetworks (*i.e. blocks*), each of which consists of one or more consecutive layers. Next, each block is attached to a small neural network called the *auxiliary network* that computes its own objective (*i.e. local loss*), from which layer weights are optimized by propagating error signals within the block. Naturally, each block can independently perform parameter updates even while other blocks process forward passes. Figure 1 illustrates the high-level overview of greedy block-wise learning.

For successful greedy block-wise learning, one must make two design decisions beforehand: (i) how to split the original network into a set of subnetworks, and (ii) which auxiliary network to use for each subnetwork. Finding the best configuration to both problems requires significant time and effort from human experts. We empirically show that the performance of greedy block-wise learning is critically sensitive to these two design choices in Appendix A. This sensitivity introduces a paradox of replacing backpropagation with greedy block-wise learning. If one has to put significant cost and time through a series of experiments to discover a workable configuration, then the benefit of greedy learning (*e.g.* reduction of training time) is diluted. Unfortunately, there has not been a generally acceptable practice to answer these two design choices.

Therefore, this work aims to propose an automated search method for the discovery of the best configuration, which has not been discussed so far. Although there have been several works on modifying backward computation graphs (Bello et al., 2017; Alber et al., 2018; Xu et al., 2018), they still rely on global end-to-end learning and focus on finding new optimizers, weight update formulas, or error propagation rules, assuming that the backward computation graphs are never discontinuous. In this work, we instead concern ourselves with making backward computation graphs discontinuous, *i.e.* finding optimal points where we stop gradient flow and use local gradients instead.

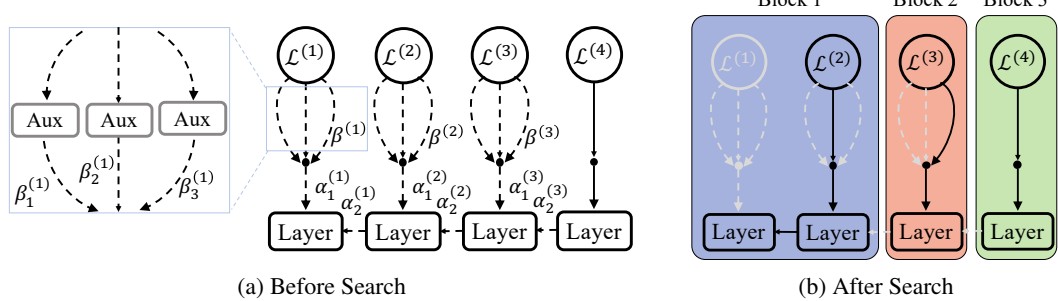

Figure 2: An illustration of our search problem when $L = 4$, $K = 3$ and $M = 3$. A search procedure should make two types of categorical choices at each layer: whether to stop the gradient flow and which auxiliary network to use. These two decision choices are represented by meta-variables $\alpha$ and $\beta$, respectively. Dashed lines represent possible decision choices, and gray ones are not chosen.

## 3    SEDONA: SEARCHING FOR DECOUPLED NEURAL ARCHITECTURES

We begin by describing the search space (Section 3.1) and present a continuous relaxation strategy that allows searching for the decoupled architecture using stochastic gradient descent on the validation loss (Section 3.2). Finally, we propose a bilevel optimization that makes SEDONA computationally efficient (Section 3.3) and the discretization on relaxed decision variables (Section 3.4).

### 3.1    THE SEARCH SPACE

Our goal is to vertically split an input network into a set of blocks to enable greedy block-wise training. This task is equivalent to solving the following two problems (Figure 2): (i) finding layers at which we discontinue the gradient flow and (ii) assigning the most appropriate auxiliary network to each of those layers for generating local error signals.

We assume that an input network consists of $L$ convolutional layers, each of which may contain normalization and pooling layers. If the network has residual connections, we regard each residual block as a single layer to simplify our notation. We focus on splitting the convolutional layers only and treat the last fully connected (FC) layers as the last block's auxiliary network.

To facilitate our exposition, we introduce two binary variables for every layer. First, the *signal variable* $\alpha^{(l)} \in \{0, 1\}$ denotes whether layer $l$ should utilize local gradients (*i.e.* the last layer of a block) or backpropagated gradients (*i.e.* the inside layer). In other words, if $\alpha^{(l)} = 1$, the layer $l$ is trained using local gradients, hence becoming the last layer in a gradient-isolated block. Otherwise, the layer is trained with backpropagated gradients, therefore becoming the inside layer in a block. Second, the *auxiliary type variable* $\beta^{(l)} \in \{0, 1\}^M$ denotes a one-hot encoding of which auxiliary is chosen out of $M$ possible candidates for layer $l$. We here assume that every layer selects its own auxiliary network; later, we leave it only for the last layer of each block. Note that the last layer of the whole network does not need these variables because the final classifier acts as its auxiliary head.

Now, we can formulate the loss for layer $l$ as follows:

$$\mathcal{L}_{train}^{(l)}(\theta, \phi, \beta) = \frac{1}{|\mathcal{D}|} \sum_{(x,y)\in\mathcal{D}} \ell^{(l)}(y, o^{(l)}) \text{ where } o^{(l)}(x;\theta) = \sum_{m=1}^{M} \beta_m^{(l)} \cdot f_m^{(l)}(a^{(l)}(x;\theta);\phi_m^{(l)}). \quad (1)$$

$\mathcal{D}$ is the training data, $\ell^{(l)}$ is the loss function, $\{f_m^{(l)}\}$ are $M$ candidates for auxiliary networks, and $a^{(l)}$ is the activations, $\theta$ is the network layer weights, and $\phi_m^{(l)}$ is the $m$-th auxiliary network weights.

The error gradients at each non-last layer $l$ are calculated as

$$\delta^{(l)} = \alpha^{(l)} \cdot \delta_{local}^{(l)} + (1 - \alpha^{(l)}) \cdot \delta_{bp}^{(l)}, \quad \text{where } \delta_{local}^{(l)} = \nabla_{a^{(l)}} \mathcal{L}_{train}^{(l)}, \quad \delta_{bp}^{(l)} = \delta^{(l+1)} \cdot \frac{\partial a^{(l+1)}}{\partial a^{(l)}}, \quad (2)$$

which is equivalent to choosing between local and backpropagated error gradients.

---

**Algorithm 1:** SEDONA – Searching for Decoupled Neural Architectures

---

Initialize signal variables $\alpha$ and auxiliary type variables $\beta$ as 0.

Pretrain layer and auxiliary network weights $(\theta, \phi)$ with Eq.(5)–(6) and store $N$ sets of weights
  with the best validation accuracies in memory $\mathcal{M}$.

**while** $(\alpha, \beta)$ *not converged* **do**

  Sample one set of layer and auxiliary network weights $(\theta, \phi)$ from $\mathcal{M}$

  Calculate $(\theta_T, \phi_T)$ by performing $T$ gradient steps on $(\theta, \phi)$ with Eq.(5)–(6).

  Update $(\alpha, \beta)$ by descending $\nabla_{(\alpha,\beta)} \mathcal{L}_{val}^{(L)}(\theta_T, \phi_T, \alpha, \beta)$ with Eq.(7).

  Calculate $(\theta_1, \phi_1)$ by performing single gradient step on $(\theta, \phi)$ with Eq.(5)–(6).

  Evict the oldest weights in $\mathcal{M}$ and save the updated weights $(\theta_1, \phi_1)$ into $\mathcal{M}$.

**end**

Obtain decoupling by discretizing $\alpha$ and $\beta$

---

## 3.2 CONTINUOUS RELAXATION

We optimize over $\alpha$ and $\beta$ after relaxing them to a continuous domain, in a similar way to DARTS (Liu et al., 2019). It enables a gradient-based optimization, which is significantly faster than directly optimizing over the discrete domain using reinforcement learning or evolutionary algorithms. This benefit accelerates especially when the search space is large, *i.e.* when the input network is deep. Henceforth, we let $\alpha^{(l)} \in \mathbb{R}^2$ and $\beta^{(l)} \in \mathbb{R}^M$. As a result, we relax the categorical choice of an auxiliary head to a weighted mixture of all possible auxiliary networks by replacing Eq.(1) with

$$\bar{o}^{(l)} = \sum_{m=1}^{M} \text{softmax}(\beta^{(l)})_m \cdot f_m^{(l)}\left(a^{(l)}; \phi_m^{(l)}\right). \tag{3}$$

Also, replacing Eq.(2), the gradients for layer $l$ become a mix of local and backpropagated gradients:

$$\bar{\delta}^{(l)} = \bar{\alpha}_1^{(l)} \cdot \delta_{local}^{(l)} + \bar{\alpha}_2^{(l)} \cdot \delta_{bp}^{(l)}, \tag{4}$$

which is the affine combination between $\delta_{local}^{(l)} = \nabla_{a^{(l)}} \mathcal{L}_{train}^{(l)}$ and $\delta_{bp}^{(l)} = \bar{\delta}^{(l+1)} \cdot \frac{\partial a^{(l+1)}}{\partial a^{(l)}}$. We use $\bar{\alpha}^{(l)} = \text{softmax}(\alpha^{(l)})$ and $\bar{\delta}^{(L)} = \nabla_{a^{(L)}} \mathcal{L}^{(L)}$.

## 3.3 BILEVEL OPTIMIZATION

We solve our search problem via bilevel optimization (Anandalingam & Friesz, 1992; Colson et al., 2007; Liu et al., 2019). In the inner level, network weights $(\theta, \phi)$ are optimized with error gradients of Eq.(4). To be specific, with fixed $(\alpha, \beta)$, we take $T$ gradient steps from initial weights $(\theta_0, \phi_0)$ for all layers $l \in \{1, \ldots, L\}$ with a learning rate $\eta$:

$$\theta_t^{(l)}(\alpha, \beta) = \theta_{t-1}^{(l)}(\alpha, \beta) - \eta \cdot \bar{\delta}^{(l)}(\alpha, \beta) \cdot \frac{\partial a^{(l)}}{\partial \theta^{(l)}}, \tag{5}$$

$$\phi_t^{(l)}(\alpha, \beta) = \phi_{t-1}^{(l)}(\alpha, \beta) - \eta \nabla_{\phi_{t-1}^{(l)}} \mathcal{L}_{train}^{(l)}\left(\theta_{t-1}, \phi_{t-1}, \alpha, \beta\right). \tag{6}$$

In the outer level, we update the meta variables $(\alpha, \beta)$ by propagating gradients with respect to the final classifier's loss $\mathcal{L}_{val}^{(L)}(\theta_T, \phi_T, \alpha, \beta)$ back through the $T$ steps:

$$\alpha \leftarrow \alpha - \eta \nabla_\alpha \mathcal{L}_{val}^{(L)}(\theta_T, \phi_T, \alpha, \beta), \qquad \beta \leftarrow \beta - \eta \nabla_\beta \mathcal{L}_{val}^{(L)}(\theta_T, \phi_T, \alpha, \beta). \tag{7}$$

Finally, we update $(\theta, \phi)$ by taking a single gradient step with updated $(\alpha, \beta)$. We repeat this process until $(\alpha, \beta)$ converge. We outline our algorithm in Algorithm 1. We empirically find that a single inner step is not sufficient to measure the effect of the meta variables on the validation loss. So, we set $T = 5$ for all experiments. Since we are only interested in $(\alpha, \beta)$ in this stage, we discard $(\theta, \phi)$ after the search. $(\theta, \phi)$ are learned again after the the final $(\alpha, \beta)$ are discretized (Section 3.4).

Additionally, we apply the following techniques for stabilizing the bilevel optimization.

**Pretraining.** Before starting the optimization of $(\alpha, \beta)$, we pretrain weights $(\theta, \phi)$ with $(\alpha, \beta)$ fixed to zeros for sufficiently long 40K iterations. This technique avoids poor evaluation of $(\alpha, \beta)$ caused

Table 1: Relative parameter and FLOP counts (%) of the auxiliary networks over the ResNet-101 on ImageNet. Our Aux $i$ heads are much more efficient than the previous MLP-SR (Belilovsky et al., 2020) and PredSim (Nøkland & Eidnes, 2019).

|  | MLP-SR | PredSim | Aux 1 | Aux 2 | Aux 3 | Aux 4 |
|---|---|---|---|---|---|---|
| Rel. FLOPs | 1.34 | 23.63 | 1.02 | 1.35 | 1.43 | 1.68 |
| Rel. # Param. | 129.33 | 23.48 | 6.38 | 7.59 | 9.68 | 12.37 |

by bad initialization of $(\theta, \phi)$. Such a warm start technique is not new, as it is adopted in Yan et al. (2019) and Chen et al. (2019). During pretraining, we store $N$ sets of $(\theta, \phi)$ with the best validation accuracies so far in memory for the purpose of weight sampling, which we will explain below. We use $N = 50$ as the size of memory.

**Weight Sampling.** In bilevel optimization, meta variables $(\alpha, \beta)$ depend on the learning trajectory of layer and auxiliary weights $(\theta, \phi)$ (*i.e.* a sequence of values of $(\theta, \phi)$ during inner optimization). As a consequence, there exists a risk of overfitting the meta variables to a specific episode (Zela et al., 2020; Chen & Hsieh, 2020). To alleviate such risk, we adopt a weight sampling scheme so that meta variables $(\alpha, \beta)$ are optimized considering various learning trajectories, not a specific one. We sample one set of weights from memory with a uniform probability at each outer optimization step, run an optimization step, and then store the updated weight back to the memory. When the memory is full, we evict the oldest one.

## 3.4 DISCRETIZATION

Equipped with $\alpha$ and $\beta$ optimized in a continuous domain, we make categorical decisions for decoupling as in Section 3.1. It is straightforward; given $K$, we split the network into $K$ blocks by selecting $K - 1$ layers with the highest learned values of $\bar{\alpha}_1$. Then, for each selected layer $l$, we choose the auxiliary head with the largest value among $\beta_1^{(l)}, ..., \beta_M^{(l)}$.

This discretization procedure shows an additional advantage of continuous relaxation. Without continuous relaxation, $K$ would be a pre-defined hyperparameter for finding a decoupling configuration; therefore, one should search for every possible $K$. On the other hand, SEDONA makes a decision based on the learned values of $\alpha$ and $\beta$; thus, it can choose any $K$ from a sorted list of $\bar{\alpha}_1$ to find the best decoupling configurations.

## 4 EXPERIMENTS

We experiment the proposed SEDONA in two stages of search and evaluation. In the search stage, SEDONA searches for the best decoupling configuration for a given neural network on CIFAR-10 to minimize the validation loss. In the evaluation stage, we split the networks according to the searched configuration, and evaluate their greedy block-wise learning performance for classification in CIFAR-10 (Krizhevsky & Hinton, 2009), Tiny-ImageNet[1] and ImageNet (Russakovsky et al., 2015). This setting will clearly show the efficiency and transferability of SEDONA since no search is required for every pair of a base network and a dataset.

### 4.1 EXPERIMENTAL SETTINGS

**Base Architectures.** We use VGG-19 and ResNet-50/101/152 as base networks, following experiments of previous literature on greedy learning (Nøkland & Eidnes, 2019; Belilovsky et al., 2020). We take the dataset complexity into account by setting the width $0.25\times$ of the original networks for CIFAR-10 and $1\times$ for Tiny-ImageNet and ImageNet. It also verifies that SEDONA can search for the configuration with a network of the reduced width on a small dataset first and apply the discovered decouplings to the ones with the original width on large datasets. We detail the base architectures in Appendix D.1

**Auxiliary Network Pool.** Reducing the size of auxiliary heads is crucial to retain the benefit of greedy block-wise learning; otherwise, the throughput gain by parallelization is diluted by the overhead of additional auxiliary heads. Thus, we heavily use depth-wise and point-wise convolution to

---

[1] http://tiny-imagenet.herokuapp.com/.

Table 2: Error rates (%) on CIFAR-10 and Tiny-ImageNet with backprop, PredSim (Nøkland & Eidnes, 2019), DGL (Belilovsky et al., 2020) and SEDONA (ours). For PredSim, DGL and SEDONA, we set $K = 4$. For CIFAR-10, we use $0.25\times$ width of the original networks.

(a) CIFAR-10

| Architecture | Backprop | PredSim | DGL | SEDONA |
|---|---|---|---|---|
| VGG-19 | 12.31 | 13.87 | 12.19 | **11.58** |
| ResNet-50 | 7.99 | 8.93 | 8.27 | **7.53** |
| ResNet-101 | 7.14 | 7.93 | 8.30 | **6.59** |
| ResNet-152 | 6.35 | 7.41 | 6.39 | **6.13** |

(b) Tiny-ImageNet

| Architecture | Backprop | PredSim | DGL | SEDONA |
|---|---|---|---|---|
| VGG-19 | 47.11 | 55.30 | 48.70 | **43.44** |
| ResNet-50 | 46.54 | 52.22 | 46.04 | **45.60** |
| ResNet-101 | 44.50 | 46.08 | 46.20 | **40.88** |
| ResNet-152 | 39.18 | 48.24 | 42.36 | **35.90** |

(a) ResNet-101

(b) ResNet-152

Figure 3: Comparison of classification errors of ResNet-101/152 when learning with increasing $K$.

let auxiliary networks in the pool be computationally lightweight. Inspired by the structure of MobileNetv2 blocks (Sandler et al., 2018), we design four auxiliary candidates called Aux 1 – 4 from the shallowest to the deepest. Aux $i$ consists of a point-wise convolution, a depth-wise convolutional layer, $(i - 1)$ inverted residual blocks and a point-wise convolution followed by an AvgPool and a FC layer. More details of auxiliary heads can be found in Appendix D.2. Table 1 shows that all four auxiliary heads have much fewer parameters and FLOPs compared to the ones from other greedy-learning methods.

**Baselines.** We compare the performance of decoupled architectures found by SEDONA with those of two state-of-the-art greedy-learning methods. We detail the optimization in Appendix E.

- **DGL** (Belilovsky et al., 2020) splits the network uniformly for a given $K$ and trains with greedy block-wise learning. Its auxiliary network is called MLP-SR, consisting of an AvgPool and 3 point-wise convolutional layers followed by an AvgPool and a 3-layer MLP.

- **PredSim** (Nøkland & Eidnes, 2019) mixes two losses, which require two different auxiliary networks, AvgPool + FC layers and a convolutional layer, respectively. Since PredSim is originally designed for greedy *layer-wise* learning, we split the network uniformly as in DGL for *block-wise* training.

## 4.2 RESULTS ON CIFAR-10 AND TINY-IMAGENET

Table 2 summarizes the classification results on CIFAR-10 and Tiny-ImageNet. DGL and PredSim perform worse than backpropagation in almost all cases. Especially, PredSim shows performance degradation on Tiny-ImageNet. Since the similarity loss of PredSim depends on supervised clustering (Nøkland & Eidnes, 2019), the local error signals could be noisy when using a large number of classes and small batch sizes. On the other hand, SEDONA enables greedy block-wise learning to outperform end-to-end backpropagation. Since the widths of the base architectures change from CIFAR-10 to Tiny-ImageNet, the results show that the decoupling found by SEDONA is successfully transferable to an architectural variant with a larger width. Interestingly, SEDONA beats backpropagation by large margins on Tiny-ImageNet, which has only 500 images per class. It implies that well-configured greedy learning could be a great learning option under data scarcity.

Figure 3 depicts the results of greedy block-wise learning with various $K$. Overall, the errors grow as $K$ increases. DGL and PredSim underperform backpropagation with large gaps when $K \geq 12$, whereas SEODNA still yields better or comparable performances to backpropagation even when $K = 16$. This result shows that SEDONA can find workable configurations for sufficiently large $K$. The quick escalation of errors with increasing $K$ hints that the block-wise training could be better than layer-wise training (*i.e.* $K = L$) for performance. Moreover, increase of $K$ significantly inflates the overhead of auxiliary heads (See Table 1 for memory and computation overheads per head).

Table 3: Error rates (%) on ImageNet with backprop, DGL and SEDONA (ours). Aux. Ens. denotes the ensemble of last two blocks' auxiliary networks.

(a) ResNet-101

| Method | $K$ | Aux. Ens. | Top-1 Err(%) | Top-5 Err(%) | Speedup |
|---|---|---|---|---|---|
| Backprop | 1 | | 21.34 | 5.86 | 1 |
| DGL | 2 | | 21.53 | 5.84 | 1.42 |
| | 4 | ✓ | 23.13 / 22.35 | 6.82 / 6.44 | 1.92 |
| SEDONA | 2 | | **20.72** | **5.39** | 1.67 |
| | 4 | ✓ | 21.32 / 21.00 | 5.83 / 5.52 | **2.01** |

(b) ResNet-152

| Method | $K$ | Aux. Ens. | Top-1 Err(%) | Top-5 Err(%) | Speedup |
|---|---|---|---|---|---|
| Backprop | 1 | | 21.22 | 5.79 | 1 |
| DGL | 2 | | 21.45 | 5.86 | 1.51 |
| | 4 | ✓ | 22.89 / 22.20 | 6.80 / 6.39 | **2.23** |
| SEDONA | 2 | | 20.69 | 5.58 | 1.61 |
| | 4 | ✓ | 21.09 / **20.20** | 5.74 / **5.13** | 2.02 |

Table 4: Comparison of error rates (%) on CIFAR-10 between (learned $\alpha$, learned $\beta$), (random $\alpha$, learned $\beta$) and (learned $\alpha$, random $\beta$) when $K = 4$.

| | ResNet-101 | | | ResNet-152 | |
|---|---|---|---|---|---|
| SEDONA | random $\alpha$ | random $\beta$ | SEDONA | random $\alpha$ | random $\beta$ |
| $6.59 \pm 0.17$ | $7.49 \pm 0.51$ | $7.07 \pm 0.37$ | $6.13 \pm 0.11$ | $7.46 \pm 0.69$ | $7.22 \pm 0.45$ |

We detail found decouplings in Appendix D.3.

### 4.3 LARGE-SCALE EVALUATION ON IMAGENET

We also evaluate the decouplings learned on CIFAR-10 on the large-scale ImageNet dataset. We present the classification errors on the validation split and training speedup in Table 3. Unlike DGL, SEDONA slightly outperforms backpropagation with $2\times$ speedup at maximum. When auxiliary networks are ensembled, SEDONA further expands the performance gaps from backpropagation. For ensembling, we sum the log-softmax outputs from the last two blocks' auxiliary headers (i.e. auxiliary headers of block $K$ and $K-1$) and use it for prediction. Note that the top-1 validation error of 20.20% by SEDONA is the state-of-the-art result in greedy block-wise learning. DGL achieves the speedup of maximum $2.23\times$, which means that the uniform split of the network may be a good policy for the throughput gain by parallelization. However, the speedup may be gained at the expense of accuracy. The reported speedup is the ratio of wall-clock training time of backprop over DGL or SEDONA. Following Huo et al. (2018a) and Belilovsky et al. (2020), we measure the speedup by spreading blocks into $K$ GPUs and do not use data parallelism.

More implementation details can be found in Appendix F.

### 4.4 ABLATION STUDIES AND DISCUSSIONS

**Effects of learned $\alpha$ and $\beta$.** SEDONA automates two types of decisions for greedy block-wise training: how to split the network and which auxiliary network to use. The signal variable $\alpha$ and the auxiliary type variable $\beta$ reflect each decision. A natural question here may be which variable is more important. To answer this, we measure the performance after modifying learned $\alpha$ or $\beta$ with random values. Table 4 describes the results. Interestingly, $\alpha$ is more critical to the performance than $\beta$, meaning that the decision for how to split is crucial for successful greedy block-wise learning.

**Importance of lower blocks.** Surprisingly, SEDONA yields much more confident values of $\bar{\alpha}_1^{(l)}$ (*i.e.* close to either 1 or 0) at lower layers than upper layers, as shown in Figure 4. It may be because learning the meta variables for upper layers is more difficult than for lower layers. However, with a sufficiently large $K$ that can split the upper layers more, the accuracy does not harm much (see Figure 3). We speculate that the difference in confidence indicates finding appropriate lower blocks is a key to success for greedy block-wise learning.

**Avoid the shallow first block.** Another interesting observation is that SEDONA prevents the first block from being too shallow, as the learned values of $\bar{\alpha}_1^{(l)}$ are smallest at several early layers (see Figure 4). It is known that the complexity of input regions that are represented by a neural network exponentially increases with its depth (Montufar et al., 2014), and thus a shallow neural network (with a finite width) may learn only too coarse features to classify complex images, which may be less meaningful representations for upper layers.

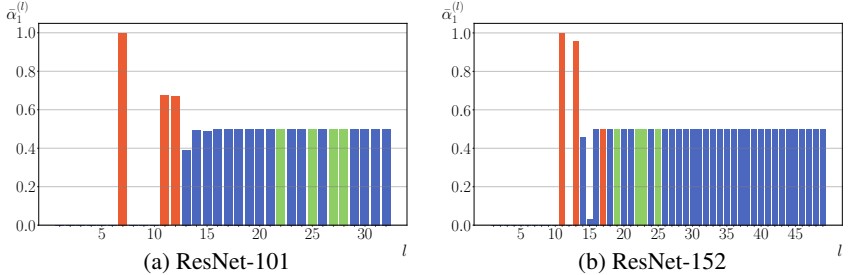

(a) ResNet-101          (b) ResNet-152

Figure 4: Learned values of $\bar{\alpha}_1^{(l)}$ on CIFAR-10. $\bar{\alpha}_1^{(l)}$ are the smallest at first few layers but large and rather invariant at upper layers. Red indicates the split layers at $K = 4$ and green indicates additionally split layers at $K = 8$.

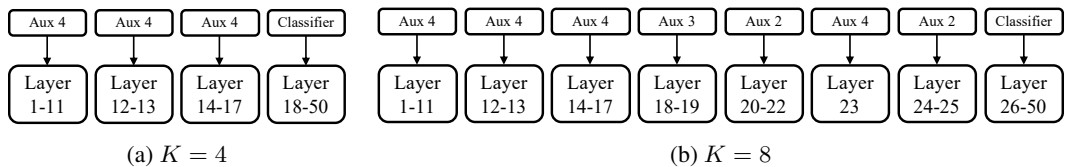

(a) $K = 4$                 (b) $K = 8$

Figure 5: Decoupled structures of ResNet-152 searched by SEDONA with $K = \{4, 8\}$.

**Use deeper auxiliary heads for lower blocks.** Figure 5 shows the decoupled structure of ResNet-152 discovered by SEDONA with $K = \{4, 8\}$. While upper layers tend to choose various auxiliary heads, lower layers prefer Aux 4, which is the deepest one in our pool. This tendency is in agreement with our intuition that a deeper auxiliary network would help lower layers learn more complex features that are more helpful for upper layers.

Additional experimental results and discussions can be found in Appendix C.

## 5 RELATED WORKS

**Greedy learning.** Greedy learning was first applied to DNNs as a pretraining step (Hinton et al., 2006). Traditionally, it solved the local optimization problems sequentially; optimization of an upper layer starts only after that of a lower layer is finished (Hinton et al., 2006; Bengio et al., 2007; Belilovsky et al., 2019). However, recent works have shown that it is possible to solve the local optimization problems jointly, with a convergence guarantee. In doing so, backward and update locking problems are alleviated (Nøkland & Eidnes, 2019; Belilovsky et al., 2020). Moreover, forward unlocking can also be achieved if each block makes use of a replay buffer (Belilovsky et al., 2020).

**Gradient approximation.** Another branch of research that tackles backward and update unlocking is gradient approximation. DNI (Jaderberg et al., 2017) trains a parametric model to output synthetic gradients from local information, removing the reliance on backpropagation to calculate true gradients. However, training with synthetic gradients often results in suboptimal accuracy, especially when neural networks are very deep. DDG (Huo et al., 2018b) and Feature Replay (Huo et al., 2018a) use time-delayed gradients or activations to ease the backward locking. Yet, they require large amounts of memory to store the activations and gradients of earlier steps. Contrarily, greedy block-wise learning achieves backward and update unlocking without storing any old information.

**Optimizing optimization.** Several works have attempted to employ automatic search methods to discover a variant of neural network optimization. Neural Optimizer Search (Bello et al., 2017) builds a new mathematical update equation from a set of primitive functions such as gradients or running average of gradients. Backprop Evolution (Alber et al., 2018) searches for a new error propagation rule, intending to improve standard backpropagation. AutoLoss (Xu et al., 2018) automatically determines the optimization schedule by selecting the loss function, scope of optimization, and duration. All of the previous works depend on continuous backward graphs. In this work, we take a different direction by splitting the backward graphs into multiple blocks.

## 6 CONCLUSION

We presented SEDONA, a novel efficient search method for finding decoupled neural architectures for successful greedy block-wise learning. Multiple state-of-the-art performances in various base networks and datasets for image classification demonstrated that the discovered architectures outperform previous human-designed greedy learning methods as well as end-to-end backpropagation. There are several interesting directions for improving SEDONA further. First, it would be interesting to automatically design local loss functions for each block, replacing the well-known cross-entropy. Second, for efficiency reasons, we used a predefined set of auxiliary heads, but searching the architectures of auxiliary networks may improve the performance and reveal each block's role, which we are yet to realize. Finally, although we assumed that the input neural network has a fixed architecture, combining SEDONA with an architecture search method such as DARTS can optimize the architecture in both forward and backward directions, which may yield intriguing results.

**Acknowledgement**. This work was supported by Samsung Advanced Institute of Technology, Institute of Information & communications Technology Planning & Evaluation (IITP) grant (No.2019-0-01082, SW StarLab) and Basic Science Research Program through National Research Foundation of Korea (NRF) funded by the Korea government (MSIT) (NRF-2020R1A2B5B03095585). Gunhee Kim is the corresponding author.

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

## A    IMPORTANCE OF BLOCK CONFIGURATIONS

As mentioned in Section 2, greedy block-wise learning is very sensitive to the choice of how to split the network into a set of blocks and the choice of the auxiliary head for each sub-network. To support our claim, we conduct a preliminary experiment depicted in Figure 6. We set $K = 4$ and randomly sample 30 configurations. For comparison, we also depict the results of backprop and SEDONA with 10 random seeds. Our experiment shows that difference in these two design choices can result in 6.01%p difference at maximum in terms of classification error rate on CIFAR-10, which is larger than the performance gap between VGG-19 and ResNet.

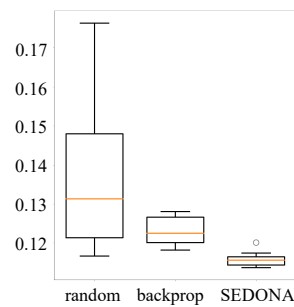

Figure 6: Classification errors of VGG-19 ($0.25\times$ width) with 30 random configurations on CIFAR-10 at $K = 4$.

## B    ADDITIONAL RELATED WORKS

**Feedback without Weight Symmetry.** Another biological implausiblity of backpropagation is that it uses the same weights for both forward and backward passes, often coined as the weight transport problem (Grossberg, 1987). A group of researches aim to overcome this problem. Target Propagation (Lee et al., 2015) uses auxiliary auto-encoders to propagate optimal activations rather than gradients. Feedback Alignment (Lillicrap et al., 2014) replaces symmetric weights of the backward pass with random weights, which is extended to a variant named Directed Feedback Alignment (Nøkland, 2016) without backward locking. However, aforementioned methods still suffer from update locking (Belilovsky et al., 2020). Moreover, there has been no result that they can yield comparable performances to backpropagation on large-scale datasets such as ImageNet. On the other hand, our work shows that greedy block-wise learning can even outperform backpropagation on ImageNet without update locking.

**Differentiable Architecture Search.** DARTS (Liu et al., 2019) provides an efficient way for neural architecture search (NAS) than previous methods based on reinforcement learning (Zoph & Le, 2017; Pham et al., 2018; Liu et al., 2018) and evolutionary algorithms (Miller et al., 1989; Angeline et al., 1994; Real et al., 2019). Moreover, it is more efficient than other gradient-based NAS methods (Luo et al., 2018; Zhang et al., 2019). However, it often fails due to overfitting to network weights, and thus several works have been proposed to overcome this limitation. Zela et al. (2020) relate the failure of DARTS with large dominant eigenvalues of $\nabla^2_\alpha \mathcal{L}_{val}$ and suggest modifying the landscape of inner optimization by adding regularization to flatten the learning landscape for outer variables. Chen & Hsieh (2020) also try to obtain smoother landscape of outer variables by injecting random or adversarial perturbations to the inner optimization objective. Our pretraining and weight sampling strategies share the motivation of these works. Rather than monitoring the Hessian or injecting noise, we regularize outer variables by taking the inner optimization over multiple weights.

## C    ADDITIONAL EXPERIMENTS AND DISCUSSIONS

### C.1    COMPARISON TO FEATURES REPLAY

We compare greedy learning methods (*i.e.* Predsim, DGL and SEDONA) to Features Replay (Huo et al., 2018a) with ResNet-101 on CIFAR-10. Features Replay suffers from more severe performance degradations when $K \geq 12$ than greedy block-wise learning methods. For Features Replay, we use the same optimization setting as in backpropagation.

### C.2    ADDITIONAL RESULTS ON IMAGENET

We also test ResNet-152 using setting of Belilovsky et al. (2020) when $K = 2$. Results can be found in Table 5. We find that the SEDONA still achieves lower test errors. However, the training setting of Belilovsky et al. (2020), which trains only for 50 epochs, does not allow backprop to fully converge, and thus the reported performance of backprop in Belilovsky et al. (2020) is significantly worse than that of ours, which trains for 600,000 iterations ($\approx 120$ epochs) as in He et al. (2016).

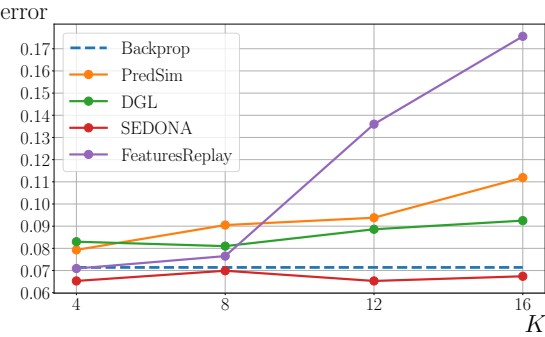

Figure 7: Comparison of classification errors of ResNet-101 when learning with increasing $K$, including Features Replay (Huo et al., 2018a).

Table 5: Error rates (%) on ImageNet with setting of Belilovsky et al. (2020) when $K = 2$. The results of Backprop and DGL are cited from Belilovsky et al. (2020).

| Method | Top-1 Error (%) | Top-5 Error (%) |
|---|---|---|
| Backprop | 25.6 | 7.9 |
| DGL | 25.5 | 8.0 |
| SEDONA | 24.95 | 7.64 |

### C.3 THE EFFECT OF DISCRETIZATION

SEDONA discretizes $(\alpha, \beta)$ after optimizing them in a continuous domain. Then, the natural question is how much performance loss the discretization procedure induces. Interestingly, we find that the discretization gap is not significant in a reasonable range of $K$. When trained on CIFAR-10, continuous versions of VGG-19 and ResNet variants achieve similar test errors compared to the discretized ones. We describe the test errors in Table 6. It may be because discretization does not affect much on both lower and upper layers for the following reasons:

1. As shown in Figure 4, the values of $\bar{\alpha}_1$ of lower layers are very close to 0 or 1, so discretization has little effect on the training of these layers.

2. For upper layers, the local error signal and the global error signal are similar in terms of their contribution to reducing the final validation loss.

   SEDONA optimizes $\alpha$, which is the relative weight of local and global gradients, to the direction that minimizes the final validation loss. So, SEDONA learns $\bar{\alpha}_1$ close to 0.5 in upper layers (Figure 4), implying that there is no significant difference between using local and global gradients for minimizing the global loss.

   One possible counterargument is that SEDONA may have learned $\bar{\alpha}_1$ close to 0.5 because it is better to mix local and global gradients with equal weights. However, this counterargument can be easily refuted by the fact that the discretization gap is still insignificant, even with a sufficiently large $K$ that can split the upper layers more (Figure 3).

### C.4 REGULARIZATION BY GREEDY LEARNING

We show that well-configured greedy learning can outperform backprop on various datasets in Section 4. As the reason of that, we conjecture that greedy block-wise learning, in general, has a regularization effect.

Table 7 shows that the training errors of DGL and SEDONA on ImageNet are smaller than that of backpropagation. However, SEDONA achieves smaller validation errors than backprop, while DGL does not. So, the decoupling configurations found by SEDONA may better control the regularization effect of greedy block-wise learning.

In fact, gradient directions with decouplings by SEDONA are more aligned to gradients by backpropagation. Figure 8 shows the average cosine similarities for each timestep interval as well as the overall cosine similarity. While both the gradients of SEDONA and DGL correlate positively with

Table 6: Error rates (%) on CIFAR-10 with Backprop, SEDONA (Cont.) and SEDONA ($K = 4$). Cont. denotes using $(\alpha, \beta)$ learned in continuous domains without discretization.

| Architecture | Backprop | SEDONA (Cont.) | SEDONA ($K = 4$) |
|---|---|---|---|
| VGG-19 | 12.31 | 12.02 | 11.58 |
| ResNet-50 | 7.99 | 7.60 | 7.53 |
| ResNet-101 | 7.14 | 6.77 | 6.59 |
| ResNet-152 | 6.35 | 6.24 | 6.13 |

Table 7: Error rates (%) on train and validation splits of ImageNet with Backprop, DGL and SEDONA. For DGL and SEDONA, we set $K = 4$.

| Architectures | Method | train error | val error |
|---|---|---|---|
| ResNet-101 | Backprop | 5.24 | 21.34 |
|  | DGL | 7.60 | 23.13 |
|  | SEDONA | 7.19 | 21.32 |
| ResNet-152 | Backprop | 4.11 | 21.22 |
|  | DGL | 5.56 | 22.89 |
|  | SEDONA | 4.13 | 21.09 |

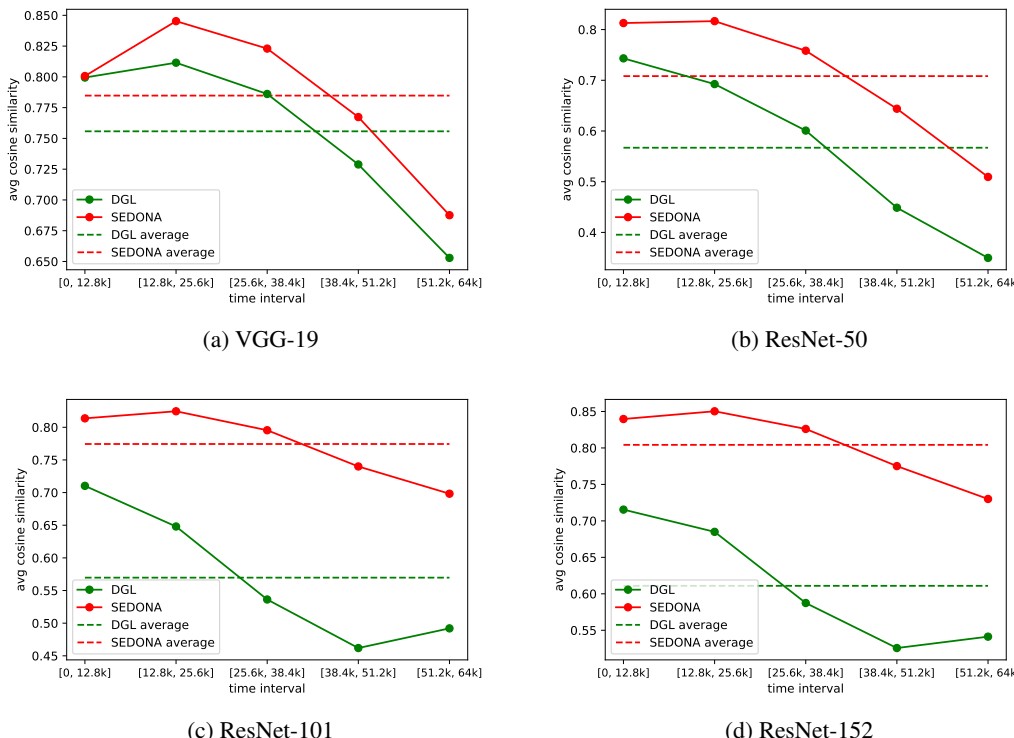

(a) VGG-19

(b) ResNet-50

(c) ResNet-101

(d) ResNet-152

Figure 8: Comparison of the average cosine similarities for each timestep interval. We compute the cosine similarity between backprop gradients and DGL/SEDONA gradients of 3x3 convolution kernel in each layer on CIFAR-10, while training with DGL and SEDONA, respectively.

the backprop gradients, the correlation in SEDONA is higher than in DGL at all time step intervals across all architectures.

One reason for appropriate regularization of SEDONA may be because SEDONA directly minimizes the final classifier's *validation loss* for optimizing $\alpha$ and $\beta$.

## C.5 THE TRADEOFF BETWEEN PERFORMANCE AND SEARCH TIME.

While decoupling configurations found by SEDONA outperforms PredSim, DGL and backprop, it requires extra search cost. The search cost could be reduced at the cost of performance by removing

Table 8: Performances and search costs of SEDONA after removing some key components one by one. Without pretraining, we 8K additional iterations for convergence.

| Method | Top-1 Error Rates (%) | | # GPUs | | Search Time (GPU days) |
|---|---|---|---|---|---|
| | CIFAR-10 | Tiny-ImageNet | Pretraining | Bilevel Opt. | |
| SEDONA ($K = 4$) | 6.59 | 40.88 | 1 | 6 | 10.1 |
| – extra inner steps | 6.62 | 41.04 | 1 | 2 | 1.3 |
| – weight sampling | 6.76 | 42.70 | 1 | 2 | 0.8 |
| – pretraining | 6.89 | 43.86 | - | 2 | 1.9 |
| Backprop | 7.14 | 44.50 | - | - | - |
| PredSim ($K = 4$) | 7.93 | 46.08 | - | - | - |
| DGL ($K = 4$) | 8.30 | 46.20 | - | - | - |

Table 9: Architecture of Aux 4 on ImageNet. For Aux $i \leq 4$, we remove final $i$ inverted residual blocks. $N_c$ denotes the number of channels at the convolutional layer on which the auxiliary head is augmented.

| |
|---|
| PwiseConv, ReLU $N_c$, BN |
| DwiseConv $3 \times 3$, ReLU $N_c$ stride 2, padding1, BN |
| InvResidual, ReLU $N_c$ stride 1, expansion ratio 4 |
| InvResidual, ReLU $1.5 \times N_c$ stride 2, expansion ratio 4 |
| InvResidual, ReLU $1.5 \times N_c$ stride 1, expansion ratio 4 |
| PwiseConv , ReLU 2048, BN |
| AdaptiveAvgPool $1 \times 1$ |
| FC 1000 |

some key components of SEDONA such as extra inner optimization steps, weight sampling and pretraining. Table 8 shows that all the components are effective for accuracy. Without both extra inner steps and weight sampling, we can still achieve the better performance than backprop with only the search cost of 0.8 GPU days.

# D  ARCHITECTURAL DETAILS

## D.1  BASE ARCHITECTURES

**ResNet-50/101/152.** For CIFAR-10 and Tiny-ImageNet, we remove the MaxPool layer after the initial convolutional layer in ResNet variants. For CIFAR-10, we also use a kernel size of 3 and a stride of 1 at the initial convolutional layer. In SEDONA, we regard each residual block in ResNets as a layer and the first convolutional layer is included in the first layer. The final AvgPool+FC layers are used as the final block's auxiliary head.

**VGG-19.** We add a batch normalization layer (Ioffe & Szegedy, 2015) after every convolutional layer, and do not use Dropout (Srivastava et al., 2014). We regard the final 3-layer MLP as the final block's auxiliary head.

## D.2  AUXILIARY HEADS

As mentioned in Section 4.1, Aux $i$ consists of a point-wise convolution, a depth-wise convolutional layer, $(i - 1)$ inverted residual blocks and a point-wise convolution followed by an AvgPool and a FC layer. We describe the detailed architectures in Table 9. For CIFAR-10, we use the $0.25\times$ width. For PredSim, we use the feature dimension of 2048 on Tiny-ImageNet and ImageNet and 512 on CIFAR-10.

Table 10: ResNet-152: 16 layers with the highest learned values of $\bar{\alpha}_1^{(l)}$ and their corresponding auxiliary headers.

| Layer Index | 11 | 13 | 17 | 19 | 22 | 23 | 25 | 26 | 28 | 29 | 49 | 31 | 48 | 38 | 37 | 36 |
|---|---|---|---|---|---|---|---|---|---|---|---|---|---|---|---|---|
| Aux. Header | 4 | 4 | 4 | 3 | 2 | 4 | 2 | 2 | 4 | 2 | 4 | 2 | 4 | 4 | 4 | 4 |

Table 11: ResNet-101: 16 layers with the highest learned values of $\bar{\alpha}_1^{(l)}$ and their corresponding auxiliary headers.

| Layer Index | 7 | 11 | 12 | 22 | 25 | 27 | 28 | 29 | 19 | 32 | 31 | 30 | 26 | 24 | 23 | 20 |
|---|---|---|---|---|---|---|---|---|---|---|---|---|---|---|---|---|
| Aux. Header | 4 | 4 | 4 | 4 | 4 | 4 | 4 | 4 | 4 | 3 | 2 | 4 | 4 | 4 | 4 | 4 |

Table 12: ResNet-50: 15 layers with the highest learned values of $\bar{\alpha}_1^{(l)}$ and their corresponding auxiliary header.

| Layer Index | 10 | 12 | 11 | 13 | 15 | 14 | 9 | 6 | 5 | 2 | 8 | 4 | 3 | 1 | 7 |
|---|---|---|---|---|---|---|---|---|---|---|---|---|---|---|---|
| Aux. Header | 4 | 4 | 4 | 4 | 4 | 1 | 1 | 2 | 1 | 1 | 1 | 1 | 1 | 1 | 1 |

Table 13: VGG-19: 15 layers with the highest learned values of $\bar{\alpha}_1^{(l)}$ and their corresponding auxiliary header.

| Layer Index | 6 | 13 | 14 | 15 | 12 | 11 | 8 | 9 | 7 | 10 | 4 | 5 | 2 | 1 | 3 |
|---|---|---|---|---|---|---|---|---|---|---|---|---|---|---|---|
| Aux. Header | 4 | 1 | 1 | 1 | 1 | 2 | 1 | 4 | 1 | 4 | 1 | 1 | 1 | 1 | 1 |

### D.3 FOUND DECOUPLINGS

Table 10 – 13 shows layers with the largest values of $\bar{\alpha}_1^{(l)}$ and their corresponding auxiliary headers with the largest value of $\beta_m^{(l)}$ for each architecture. For ResNet-101/152, we show 16 layers with largest values of $\bar{\alpha}_1^{(l)}$ for better readability.

## E OPTIMIZATION DETAILS

### E.1 OPTIMIZATION AT SEARCH STAGE

We search for decouplings for VGG-19 and ResNet-50/101/152 on CIFAR-10. We use 40% of CIFAR-10 training split as a validation set. For the outer optimization, we use Adam optimizer (Kingma & Ba, 2015) with a fixed learning rate of 0.01 and a weight decay of 0.000001. For the inner optimization, we use SGD with a momentum of 0.9 and a weight decay of 0.001. We use an initial learning rate of 0.1 and decay it down to 0.001 with the cosine annealing learning rate decay (Loshchilov & Hutter, 2017). Label smoothing (Szegedy et al., 2016) of 0.1 is also used. We repeat bilevel optimization steps for 2K iterations. As mentioned in Section 3.3, we pretrain weights for 40K iterations with outer variables fixed as zero and store 50 sets of weights with the best validation accuracies. For pretraining, we use the same setting as in the inner optimization. During pretraining, we evaluate weights on the validation set at every 80 iterations. For pretraining and the inner optimization, we use the standard data augmentation strategy of padding, random crop and random flip.

### E.2 OPTIMIZATION AT EVALUATION STAGE

In all experiments, we use the cosine annealing learning rate decay (Loshchilov & Hutter, 2017). Label smoothing (Szegedy et al., 2016) of 0.1 is used for Backprop, DGL and SEDONA. For PredSim, 0.99 is used as a coefficient of similarity loss as in Nøkland & Eidnes (2019).

**CIFAR-10.** For evaluation on CIFAR-10 dataset, we train all our models for 64K iterations with a batch size of 128 using SGD with a momentum of 0.9. For all experiments on CIFAR-10, we use an initial learning rate of 0.1 and decay it down to 0.001. For VGG-19 and ResNet-50/101, we use a weight decay of 0.0001 while a weight decay of 0.0002 is used for ResNet-152. 10% of train data

Table 14: Search costs of SEDONA.

| Architecture | # GPUs | | Search Time | | |
| --- | --- | --- | --- | --- | --- |
| | Pretraining | Bilevel Opt. | Pretraining (days) | Bilevel Opt. (days) | Total (GPU days) |
| VGG-19 | 1 | 4 | 0.1 | 0.5 | 2.1 |
| ResNet-50 | 1 | 4 | 0.2 | 0.9 | 3.8 |
| ResNet-101 | 1 | 6 | 0.3 | 1.6 | 10.1 |
| ResNet-152 | 1 | 6 | 0.5 | 2.5 | 15.5 |

Table 15: Wall-clock training times (hours) of Backprop, DGL and SEDONA (ours) on ImageNet .

| Architecture | Backprop | DGL | | SEDONA | |
| --- | --- | --- | --- | --- | --- |
| | | $K = 2$ | $K = 4$ | $K = 2$ | $K = 4$ |
| ResNet-101 | 113.5 | 80.2 | 59.2 | 67.3 | 56.5 |
| ResNet-152 | 158.5 | 104.8 | 71.2 | 98.5 | 78.3 |

is used as the validation set. We follow the standard data augmentation strategy of padding, random crop and random flip as in Nøkland & Eidnes (2019).

**Tiny-ImageNet.** For evaluation on Tiny-ImageNet dataset, we train all our models for 30K iterations with a batch size of 256. For Backprop, DGL and SEDONA, an initial learning rate is 0.1, which is decayed down to 0.001 with cosine annealing, and SGD with a momentum of 0.9 is used. A weight decay of 0.0001 is used for VGG-19, ResNet-50/101, and a weight decay of 0.0002 is used for ResNet-152. However, we fail to train well with the momentum optimizer for PredSim. So, we use Adam optimizer (Kingma & Ba, 2015) instead. We use an initial learning rate of 0.0005, which is decayed down to 0.000001, and a weight decay of 0.0001 for all architectures for PredSim. In addition, we subsample 40 classes per batch until 15K iterations following Nøkland & Eidnes (2019). We use the standard data augmentation strategy of padding, random crop, random rotation and random flip.

**ImageNet.** For ImageNet dataset, we train all our models for 600K iterations with a batch size of 256 using SGD with a momentum of 0.9. A weight decay of 0.00005 is used for both ResNet-101 and ResNet-152. We use an initial learning rate of 0.1 decayed down to 0.0001 for ResNet-101, and 0.05 down to 0.00001 for ResNet-152. We use the standard data augmentation of random resized crop and random flip. We report the accuracy in the single-crop setting.

# F   IMPLEMENTATION & COMPUTING ENVIRONMENTS

## F.1   IMPLEMENTATION DETAILS

For implementation, we use Python 3.8 and PyTorch 1.6.0. At the search stage, we use the higher library[1] to enable differentiable weight updates in PyTorch computational graphs. For evaluation, we implement asynchronous updates of blocks by introducing queues between blocks. For PredSim, DGL and Features Replay implementations, we refer to their official PyTorch implementations[2]. We use mixed precision training with Apex[3] on Tiny-ImageNet and ImageNet.

## F.2   COMPUTING ENVIRONMENTS

We report our search cost and wall-clock training time in Table 14 and 15, respectively. All experiments are conducted with total 8 NVIDIA Quadro 6000 GPU cards and 2 8-core Intel Xeon E5-2620 v4 processors with 256 GB RAM. For search, we use 4 GPUs for VGG-19 and ResNet-50 and 6 GPUs for ResNet-101/152.

---

[1]https://github.com/facebookresearch/higher.

[2]PredSim: https://github.com/anokland/local-loss, DGL: https://github.com/eugenium/DGL, Features Replay: https://github.com/slowbull/FeaturesReplay.

[3]https://github.com/NVIDIA/apex

