# OpenReview forum: "SEDONA: Search for Decoupled Neural Networks toward Greedy Block-wise Learning"
_ICLR.cc/2021/Conference — ICLR 2021 Poster_

### Official Review · AnonReviewer4 · 2020-10-23
**Major concern is about the experimental results**

**Rating:** 7
**Confidence:** 4

**Review:**

Summary Of Contributions: This paper proposes to automate the design of auxiliary network and its allocation under decoupled neural network scheme, a design that speeds up network training and potentially boost model accuracy. The approach is validated with leading performance on ResNet and VGGNet under various datasets.


Strengths:  The idea to search auxiliary network for decoupled neural network is novel, and the proposed method is verified on multiple widely used datasets.

Weaknesses:

1.	It’s unclear if the proposed differentiable search algorithm works better than DARTs with direction comparison, as the proposed method is basically DARTs with some changes (e.g. weight sampling, extra inner optimization steps per outer optimization), that are also applied to typical NAS tasks, a comparison with DARTs is needed to validate how much of the gain is from the proposed method rather than the fact of automating network design itself.

2.	It seems that the decoupled neural network scheme has nearly no gains in terms of Top-1 without the auxiliary heads ensembled as shown in Table 3. The details should be provided on how the ensemble works, are ensembled auxiliary heads kept during the testing stage? If so, Table 3 cannot tell if the gain is from the ensembled aux heads or decoupled neural network scheme. Network trained with Backprop method and identical extra aux head for ensemble might reach the same level of performance.

3.	Experiment with DGL in Table3 (b) is problematic, DGL in its original paper only provides ResNet-152 result with K=2, and it is better than Backprop according to their experiment. Yet the authors set K=4 in Table 3 and got a contrary result. This brought up the suspect of cherry picking of hyperparameters and can only be addressed with stricter comparison.

4.	One of the advantages to adopt a decoupled neural network scheme is training speedup, while the network based on compared methods is instantly available, the proposed method requires extra time to search before training. With this extra cost, how much the training speedup benefit is still left? The search cost should be indicated and the overall time cost should be discussed.

---

> ### Author Response · Authors · 2020-11-18
> **Response to Reviewer 4 [Part 2/2]**
>
> **5. (Q4) While the network based on compared methods is instantly available, the proposed method requires extra time to search before training.**
>
> **A.** The review is correct that the compared methods do not require additional search steps.  However, these methods yield worse performance than backprop. Moreover, their performances become worse as $K$ increases (Figure 3), so the speedup is also limited in practice due to large sacrifices of accuracy. On the other hand, the decoupling configurations found by SEDONA do not suffer from these issues. SEDONA shows consistently similar or better performance with multiple off-the-shelf CNNs in multiple datasets compared to backprop. Moreover, SEDONA still performs similar or better than backprop up to $K=16$ (Figure 3).
>
> **6. (Q4) The search cost should be indicated.**
>
> **A.** Search time of SEDONA is 0.6, 1.1, 1.9 and 3 days for VGG-19, ResNet-50, ResNet-101 and ResNet-152, respectively, as described in Table 14 in Appendix (Table 10 in the old draft).
>
> **7. (Q4) With this extra cost, how much of the training speedup benefit is still left? The overall time cost should be discussed.**
>
> **A.** Although SEDONA requires additional search time, the search stage is performed only once per network architecture. No additional search is required no matter whether the dataset changes or the number of desired blocks $K$ changes. Moreover, since training times of ResNet-152 on ImageNet using backprop and SEDONA were 158.5 hours and 78.3 hours, respectively, the search cost is canceled out even when trained once.
>
>
> **References**
>
> [1] Eugene Belilovsky, Michael Eickenberg, and Edouard Oyallon. Greedy layerwise learning can scale to imagenet. In ICML, 2019.
>
> [2] Eugene Belilovsky, Michael Eickenberg, and Edouard Oyallon. Decoupled greedy learning of cnns. In ICML, 2020.
>
> [3] Kaiming He, Xiangyu Zhang, Shaoqing Ren, and Jian Sun. Deep residual learning for image recognition. In CVPR, 2016.

---

> ### Author Response · Authors · 2020-11-18
> **Response to Reviewer 4 [Part 1/2]**
>
> Thank you for your valuable and detailed review. We will clarify all the concerns in our final draft and make our code public.
>
> **1. (Q1) Comparison with DARTs.**
>
> **A.** For comparison, we ran an ablation study by subtracting key components of our method one by one: (1) without extra inner optimization steps (i.e. $T=1$), (2) without weight sampling, and (3) without pretraining, which is equivalent to DARTS. Table below shows the top-1 error rate (%) of decoupling configurations found for ResNet-101, indicating that each of our components is effective and SEDONA is better than DARTS.
>
> |                        | CIFAR-10 | Tiny-ImageNet |
> |------------------------|:--------:|:-------------:|
> | SEDONA                 | 6.59     | 40.88         |
> | - extra inner steps    | 6.62     | 41.04         |
> | - weight sampling      | 6.76     | 42.70         |
> | - pretraining (=DARTS) | 6.89     | 43.86         |
>
>
> **2. (Q2) The details should be provided on how the ensemble works.**
>
> **A.** For ensembling, we sum the log-softmax outputs from the last two blocks’ auxiliary headers (i.e. auxiliary headers of block $K$ and $K-1$) and use it for prediction. We will add this to the draft.
>
> **3. (Q2) Table 3 cannot tell if the gain is from the ensembled aux heads or decoupled neural network scheme.**
>
> **A.** As the reviewer suggested, we measure the performance of backprop and DGL when the ensemble is applied. We found that they do not attain the same level of performance as SEDONA.
>
> |                    | Top-1 Error (%) | Top-5 Error (%) |
> |--------------------|:---------------:|:---------------:|
> | BP (Aux. Ens.)     | 20.94           | 5.77            |
> | DGL (Aux. Ens.)    | 22.20           | 6.39            |
> | SEDONA (Aux. Ens.) | 20.20           | 5.13            |
>
>
> We believe that ensembling is one benefit of greedy block-wise learning over backprop since greedy block-wise learning naturally supports multiple auxiliary heads [1]. In the above experiments, we add an auxiliary head to backprop in the same way with SEDONA but it may be rather artificial and the performance improvement by ensembling in backprop is far less than greedy learning methods (DGL and SEDONA).
>
>
> **4. (Q3) Performance of ResNet-152 on ImageNet when $K=2$.**
>
> **A.** We chose $K=4$ in ImageNet experiments because speedup is significant when $K=4$ and more splitting is encouraged for the purpose of greedy block-wise learning. As the review suggested, we evaluate ResNet-152 with K=2, where SEDONA still performs better than Backprop and DGL.
>
> |              | Top-1 error (%) | Top-5 error (%) | Speedup |
> |--------------|:---------------:|:---------------:|:-------:|
> | Backprop     | 21.22           | 5.79            | 1       |
> | DGL (K=2)    | 21.45           | 5.86            | 1.51    |
> | SEDONA (K=2) | 20.69           | 5.58            | 1.61    |
>
> In addition, note that the training setting of [2], which trains only for 50 epochs, did not allow backprop to fully converge, and thus the reported performance of backprop in DGL is significantly worse than that of ours, which trains for 600,000 iterations ($\approx 120$ epochs) as in the original ResNet paper [3]. We also tested ResNet-152 using [2]’s setting when $K=2$, and found that the SEDONA still achieves lower test errors. The results of Backprop and DGL are cited from [2].
>
> |             | Top-1 error (%) | Top-5 error (%) |
> |-------------|:---------------:|:---------------:|
> | Backprop    | 25.6            | 7.9             |
> | DGL(K=2)    | 25.5            | 8.0             |
> | SEDONA(K=2) | 24.95           | 7.64            |

---

### Official Review · AnonReviewer2 · 2020-10-26

**Rating:** 7
**Confidence:** 4

**Review:**

The paper proposes a method for decoupled training of neural networks called SEDONA. In the spirit of recent trends in greedy layer-wise and indirect training, SEDONA allows gradient information to flow either from the next layer as in backpropagation or from an auxiliary head, trying to make a prediction using the current layer's output. Since a direct search for the best decoupled configuration results into probing a combinatorial number of splits, authors propose a continuously relaxed formulation which they later discretize. Transferring the found decoupled configurations to datasets different from the "pretraining" ones shows improvements in terms of validation accuracy and training time.

I find the overall topic of asynchronous or backpropagation-free training of neural networks very interesting and the submitted paper particularly relevant to this topic. The proposed search algorithm is novel to my knowledge and may be found useful in the community. However, I think authors should have put more effort into understanding SEDONA.

The major points are the following:
1. The effect of discretization is not studied. I would be interested in seeing how well does a continuous configuration perform and what kind of accuracy loss (if any) is caused by discretization.
2. I wonder why SEDONA performs better than backprogation (sometimes much better as in Tiny ImageNet)? Is it because not following the true (stochastic) gradient acts as regularization? One way of assessing this would be to also provide training loss or accuracy values. Why is your top1 error value for ResNet-152 is about two percent higher that in the original paper by He et al, 2015?
3. If the update directions configured by SEDONA lead to such a good performance, how do they correspond to the true gradients? Do they correlate positively? To what extent can such decoupled feedback implement credit assignment in a neural network?

I believe that addressing these questions is very important prior to publication of the method.

Minor comments:
1. $\theta$ does not appear on the RHS of (1).
2. Auxiliary head ensembling is not explained very clearly.

---

> ### Author Response · Authors · 2020-11-18
> **Response to Reviewer 2 [Part 2/2]**
>
> **4. (Q3) How do the update direction of SEDONA correspond to the true gradients? Do they correlate positively?**
>
> **A.** We found that both the gradients of SEDONA and DGL correlate positively with the true (backprop) gradients. To measure the correlation, we computed the cosine similarity between backprop gradients and SEDONA/DGL gradients of 3x3 convolution kernel in each layer on CIFAR-10. Below we show the average cosine similarities for each timestep interval as well as the overall cosine similarity. However, the correlation in SEDONA is higher than in DGL at all time step intervals across all architectures.
>
> | (VGG-19)&nbsp;&nbsp;&nbsp;&nbsp;&nbsp;&nbsp;&nbsp;&nbsp;| &nbsp;(0,12.8K]&nbsp; |&nbsp; (12.8K,25.6K]&nbsp; |&nbsp; (25.6K,38.4K]&nbsp; |&nbsp; (38.4K,51.2K] &nbsp;|&nbsp; (51.2K,64K]&nbsp; |&nbsp; Overall&nbsp; |
> |:----------|:-----------:|:---------------:|:---------------:|:---------------:|:-------------:|:---------:|
> | DGL      | 0.7995    | 0.8115        | 0.7861        | 0.7288        | 0.6529      | 0.7558  |
> | SEDONA   | 0.8007    | 0.8453        | 0.8230        | 0.7673        | 0.6876      | 0.7848  |
>
> | (ResNet-50)&nbsp;&nbsp; | &nbsp;(0,12.8K]&nbsp; |&nbsp; (12.8K,25.6K]&nbsp; |&nbsp; (25.6K,38.4K]&nbsp; |&nbsp; (38.4K,51.2K] &nbsp;|&nbsp; (51.2K,64K]&nbsp; |&nbsp; Overall&nbsp; |
> |:----------|:-----------:|:---------------:|:---------------:|:---------------:|:-------------:|:---------:|
> | DGL         | 0.7432    | 0.6925        | 0.6008        | 0.4487        | 0.3497      | 0.5670  |
> | SEDONA      | 0.8128    | 0.8166        | 0.7583        | 0.6439        | 0.5094      | 0.7082  |
>
> | (ResNet-101) |&nbsp;(0,12.8K]&nbsp; |&nbsp; (12.8K,25.6K]&nbsp; |&nbsp; (25.6K,38.4K]&nbsp; |&nbsp; (38.4K,51.2K] &nbsp;|&nbsp; (51.2K,64K]&nbsp; |&nbsp; Overall&nbsp; |
> |:----------|:-----------:|:---------------:|:---------------:|:---------------:|:-------------:|:---------:|
> | DGL          | 0.7103    | 0.6480        | 0.5363        | 0.4619        | 0.4920      | 0.5697  |
> | SEDONA       | 0.8137    | 0.8245        | 0.7955        | 0.7400        | 0.6983      | 0.7744  |
>
> | (ResNet-152) | &nbsp;(0,12.8K]&nbsp; |&nbsp; (12.8K,25.6K]&nbsp; |&nbsp; (25.6K,38.4K]&nbsp; |&nbsp; (38.4K,51.2K] &nbsp;|&nbsp; (51.2K,64K]&nbsp; |&nbsp; Overall&nbsp; |
> |:----------|:-----------:|:---------------:|:---------------:|:---------------:|:-------------:|:---------:|
> | DGL          | 0.7155    | 0.6850        | 0.5874        | 0.5257        | 0.5412      | 0.6109  |
> | SEDONA       | 0.8397    | 0.8503        | 0.8260        | 0.7751        | 0.7300      | 0.8042  |
>
> **5. (C1) $\theta$ does not appear on the RHS of (1).**
>
> **A.** Thank you for pointing out missing $\theta$. We will update Equation (1) accordingly.
>
> **6. (C2) Details on auxiliary head ensemble**
>
> **A.** For ensembling, we sum the log-softmax outputs from the last two blocks’ auxiliary headers (i.e. auxiliary headers of block $K$ and $K-1$) and use it for prediction. We will add this to the draft.
>
>
> **References**
>
> [1] Kaiming He, Xiangyu Zhang, Shaoqing Ren, and Jian Sun. Deep residual learning for image recognition. In CVPR, 2016.
>
> [2] https://pytorch.org/hub/pytorch_vision_resnet

---

> > ### Comment · AnonReviewer2 · 2020-11-25
> > **Review revision**
> >
> > Thank you for the detailed reply. I'm mostly satisfied with the provided answers but really wish my question regarding credit assignment capabilities of SEDONA was addressed. To re-iterate, it's not that I find the results underwhelming, it's that I would like more analysis and understanding of why they are so great. Assuming authors will incorporate all these additional information in the paper, I'm willing to raise my score.

---

> > > ### Author Response · Authors · 2020-11-25
> > > **Thank you for examining our revisions**
> > >
> > > We sincerely appreciate spending your time examining our responses and revisions. We also thank you for your valuable suggestions and comments. We will further investigate the credit assignment of greedy learning.

---

> ### Author Response · Authors · 2020-11-18
> **Response to Reviewer 2 [Part 1/2]**
>
> Thank you for your valuable and detailed review. We will clarify all the concerns in our final draft and make our code public.
>
> **1. (Q1) The effect of discretization should be studied.**
>
> **A.** We found that the discretization gap is not significant in a reasonable range of $K$. When trained on CIFAR-10, the continuous version of ResNet-101 achieved the test error of 6.77%, which is not different from the discretized one, whose test errors were 6.59%, 6.99%, 6.53%, 6.74% for $K=4, 8, 12, 16$, respectively. It may be because discretization does not affect much on both lower and upper layers for the following reasons:
>
> 1. As shown in Figure 4, the values of $\bar{\alpha}_1$ of lower layers are very close to 0 or 1, so discretization has little effect on the training of these layers,
> 2. For upper layers, the local error signal and the global error signal are similar in terms of their contribution to reducing the final validation loss.
> SEDONA optimizes $\alpha$, which is the relative weight of local and global gradients, to the direction that minimizes the final validation loss (Equation (7)). So, SEDONA learns $\bar{\alpha}_1$ close to 0.5 in upper layers (Figure 4), implying that there is no significant difference between using local and global gradients for minimizing the global loss.
> One possible counterargument is that SEDONA may have learned $\bar{\alpha}_1$ close to 0.5 because it is better to mix local and global gradients with equal weights. However, this counterargument can be easily refuted by the fact that the discretization gap is still insignificant, even with a sufficiently large $K$ (the number of blocks) that can split the upper layers more.
>
> **2. (Q2) Why is your top1 error value for ResNet-152 is about two percent higher than that in the original paper?**
>
> **A.** Table 3 and 4 in [1] report the test error in 10-crop testing and multiple scale testing, respectively. We report the test error in 1-crop setting as in the official PyTorch repository [2]. The top-1 test error for ResNet-152 using backpropagation is 21.22% in our paper, which is slightly better than 21.69% reported in the official PyTorch official repository [2].
>
> **3. (Q2) Why does SEDONA perform better than backpropagation?**
>
> **A.** As the review mentioned, we conjecture that greedy block-wise learning, in general, has regularization effects. One evidence is that when training a very deep network with a small dataset, the increasing depth of the network does not harm the performance in greedy block-wise learning. For instance, Table 6 in [1] shows that ResNet-1202 performs worse than ResNet-110 on CIFAR10 when trained using backpropagation. The authors of [1] argue that such a phenomenon is due to overfitting of ResNet-1202. However, we found that when trained with greedy block-wise learning (DGL, to be specific), the increased depth of ResNet-1202 improves the performance. Table below shows the test error of ResNet-110 and ResNet-1202 on CIFAR-10. (The backprop result is cited from [1])
>
> |          | ResNet-110 | ResNet-1202 |
> |----------|:----------:|:-----------:|
> | Backprop |     6.43   |      7.93   |
> | DGL      |     8.60   |      7.36   |
>
> Moreover, the table below shows that the training errors of DGL and SEDONA on ImageNet are indeed smaller than that of backpropagation.
>
> | (ResNet-101) | train error | val error |&nbsp;&nbsp;&nbsp;&nbsp;    | (ResNet-152) | train error | val error |
> |--------------|:-----------:|:---------:|---|:-------------|:-----------:|:---------:|
> | Backprop     | 5.24        | 21.34     |   | Backprop     | 4.11        | 21.22     |
> | DGL          | 7.6         | 23.13     |   | DGL          | 5.56        | 22.89     |
> | SEDONA       | 7.19        | 21.32     |   | SEDONA       | 4.13        | 21.09     |
>
> SEDONA achieves smaller validation errors than backprop, while DGL does not. So, the decoupling configurations found by SEDONA may better control the regularization effect of greedy block-wise learning. One reason may be because SEDONA directly minimizes the final classifier’s “validation” loss for optimizing $\alpha$ and $\beta$. We found another evidence supporting this conjecture, but it is related to the gradient directions of SEDONA and DGL, so we will describe it in the following answer to Q3.

---

### Official Review · AnonReviewer1 · 2020-10-28
**Constrained architecture search that seems to mildly improve performance and speed up training by 2x, but somewhat unclear and hard to understand**

**Rating:** 6
**Confidence:** 3

**Review:**

### Summary

This paper proposes a differentiable architecture search approach for splitting a deep network into locally-trained blocks to achieve training speedup. The approach achieves better performance than using backprop on small datasets (CIFAR10 and TinyImageNet), and comparable or slightly improved performance on ImageNet with 2x claimed training speedup. Learned network architecture choices seem to transfer between datasets.

### Strong points

S1: The method outperforms comparable baselines, and significantly improves performance over standard backprop for small datasets (CIFAR10 and TinyImageNet)

S2: The learned network architectures transfer between tasks.

S3: On ImageNet, the method achieves slightly improved performance with claimed 2x training speedup.

### Weak points

W1: This seems like a pretty complicated approach to only get 2x speedup in training, and besides slight improvement in performance, that seems like the only benefit this method achieves for realistically large datasets like ImageNet.

W2: "Speedup" is not really defined. i assume this is the wall-clock time to achieve convergence, but I couldn't find a definition of convergence or stopping criterion, or how speedup is measured.

W3: Complicated optimization tricks seem necessary to get the bilevel optimization to work (Section 3.3: "In bilevel optimization, meta variables (α, β) depend on the learning trajectory of layer and auxiliary weights (θ, φ) (i.e. a sequence of values of (θ, φ) during inner optimization). As a consequence, there exists a risk of overfitting the meta variables to a specific episode"). If I have the choice between 2x training time with straight-forward optimization, versus 1x training time with complicated bilevel optimization that needs to be tuned, I would probably choose the former option, since I need to spend a lot less time debugging and tuning parameters.

### Recommendation

I think this paper is marginally below the acceptance threshold. The results seem decent, though a little underwhelming (see W1), but the biggest concern for me is that key concepts are unclear or ambiguous (e.g. definition of speedup is unclear, see W2, "confidence of \alpha at lower layers", see C2, and how the "backprop" gradients are computed and used in equation (2), see Q2).

If these issues were clarified I would be inclined to increase my score. But as it stands, this feels like a complicated method without much payoff (only 2x training speedup with slightly improved performance). It's possible that I'm missing something or not understanding the potential impact of the results, and explicit discussion of this and/or improved clarity of the description and evaluation of the method may help.

### Questions

Q1: What is the absolute wall-clock time for training that the speedup is reported on?

Q2: Given that the gradients used to train each block are a combination of local and backpropped gradients, how does this method avoid the locking problem, and thus provide speedup?

### Other comments

C1: The "2.02x" speedup number looks a little funny; are two decimal places justified? Perhaps better to just say "2x speedup"?

C2: In Section 4.4, the paper says, "Surprisingly, SEDONA yields much more confident values of $\alpha_1^{(l)}$ (i.e. close to either 1 or 0) at lower layers than upper layers,", but when I look at Figure 4, at first glance I thought that \alpha is _less_ confident at lower iterations, according to the provided definition, compared to later iterations (which are all at 1). But then I realized the plot uses a log scale. Consider using a linear scale here instead? Or at least describe the confidence in terms of the log scale.

C3: I find the usage of all-caps CONV to be a little odd, at least I've never seem that before. Maybe just say "convolution"? Doesn't seem like CONV saves much space.

C4: I find this key statement to be unclear: "denotes whether layer l should utilize local gradients (i.e. the last layer of a block) or backpropagated gradients (i.e. the inside layer)." Is this saying that if a local gradient is used for a layer, then it is a candidate for being the last layer of a block, and if backpropped gradient is used, then it is a candidate for being the inside layer of a block? Some rewording of this sentence should help.

C5: Typos, in Appendix B: "to flat the learning landscape" -> "to flatten the learning landscape", Section 4.1: "to let auxiliary networks in the pool computationally lightweight" -> "to let auxiliary networks in the pool be computationally lightweight", Section 3.3: "Such warm start" -> "Such a warm start", Section 1: "If local signals are lousy representative of the global goal" -> "If local signals are not representative of the global goal"

---

> ### Author Response · Authors · 2020-11-18
> **Response to Reviewer 1 [Part 2/2]**
>
> **5. (C1) Perhaps better to just say "2x speedup"?**
>
> **A.** Thanks for the suggestion. We will update the draft soon.
>
> **6. (C2) When I look at Figure 4, at first glance I thought that \alpha is less confident at lower iterations, according to the provided definition, compared to later iterations (which are all at 1). Consider using a linear scale here instead?**
>
> **A.** We initially chose the log scale because values of $\bar{\alpha}^{(l)}_{1}$ at first few layers are too close to 0, and were not visible in the linear scale plot. However, we acknowledge that using log scale made it more difficult to understand the notion of confidence. Following your feedback, we will revise Figure 4 using a linear scale.
>
> **7. (C3) Maybe just say "convolution"?**
>
> **A.** We will replace “CONV” with “convolution” in the final draft.
>
> **8. (C4) The key statement seems to be unclear: "denotes whether layer l should utilize local gradients (i.e. the last layer of a block) or backpropagated gradients (i.e. the inside layer)."**
>
> **A.** Your understanding is correct. The structure of blocks can be expressed using signal variables $\alpha^{(l)}$. If $\alpha^{(l)}=1$, the layer $l$ is trained using local gradients, therefore becoming the last layer in a gradient-isolated block. If $\alpha^{(l)}=0$, the layer $l$ is trained using gradients backpropagated from its upper layer within a block, therefore becoming an inside layer of a gradient-isolated block (see Equation (2)). We will rewrite the sentences clearly.
>
> **9. (C5) Typos**
>
> **A.** Thank you for pointing out typos, which will be corrected.
>
> **References**
>
> [1] Eugene Belilovsky, Michael Eickenberg, and Edouard Oyallon. Decoupled greedy learning of cnns. In ICML, 2020.
>
> [2] https://pytorch.org/hub/pytorch_vision_resnet
>
> [3] Zhouyuan Huo, Bin Gu, Qian Yang, and Heng Huang. Decoupled parallel backpropagation with convergence guarantee. In ICML, 2018.
>
> [4] Kaiming He, Xiangyu Zhang, Shaoqing Ren, and Jian Sun. Deep residual learning for image recognition. In CVPR, 2016.

---

> > ### Comment · AnonReviewer1 · 2020-11-24
> > **Updated review**
> >
> > Thanks to the authors for the responses and for their additional work to improve the paper. I read through the paper revisions the author responses. These responses and the associated revisions helped clear up a lot of my concerns or confusions, and thus I increased my score a bit.
> >
> > However, I still have some concern about the applicability of the proposed method. The authors use a restricted class of convolutional architectures, and claim that once their methods finds a decoupled architecture, it can be applied to other datasets. But I think it's often that case that deep learning practitioners want to try a variety of architectures, and it sounds like the initial search stage needs to be run for any new architecture that is being used; furthermore, this search stage can be significant in terms of wall clock time. Thus, it seems a bit optimistic to claim that the search stage only needs to be done once per architecture, and to discount the time it takes to run the search stage.
> >
> > Furthermore, I'm a bit skeptical that dataset-invariance is a general property of this approach, as the authors only showed this across datasets for the same task of image classification. I kind of doubt that the architecture search would generalize across tasks, e.g. an architecture tuned for image classification may not generalize well to, say, audio classification. I think some discussion of these issues in a future revision would improve the paper.

---

> > > ### Author Response · Authors · 2020-11-25
> > > **Thank you for examining our revisions**
> > >
> > > We sincerely appreciate spending your time examining our responses and revisions. We also thank you for your valuable suggestions and comments.
> > >
> > > Although we did not claim transferability across tasks with different modalities, checking it seems to be interesting and we will try to evaluate found decouplings on other tasks. We agree that reducing the search cost would be helpful. To alleviate the search cost, practitioners might omit taking extra inner steps, which is the most time-consuming but least significant part of our algorithm. So, we will provide a discussion on the tradeoff between search efficiency and performance in the final draft.

---

> ### Author Response · Authors · 2020-11-18
> **Response to Reviewer 1 [Part 1/2]**
>
> Thank you for your valuable and detailed review. We will clarify all the concerns in our final draft and make our code public.
>
> **1. (W1) This seems like a pretty complicated approach to only get 2x speedup in training**
>
> **A.** The main objective of greedy (block-wise) learning is to solve update/backward locking problems of backpropagation. Given that optimization of almost all neural networks solely relies on backprop, seeking another workable optimization strategy beyond backprop is a critical topic for deep learning research. As one of the most promising alternatives, greedy learning has been studied much but its performance has been highly limited compared to backprop. For example, the previous state-of-the-art greedy learning method achieved 25.5% Top-1 error rate with ResNet-152 [1], which is about 4%p higher than that of the pretrained ResNet-152 in the official PyTorch repository [2]. The key contribution of SEDONA is that we can automatically find the best performing configuration of greedy learning that not only outperforms existing greedy learning methods but also similar or better compared to backprop. 2x speedup is one beneficial byproduct (not the primary goal) of our approach.
>
>
> **2. (W2, Q1) "Speedup" is not really defined.**
>
> **A.** We reported speedup as the ratio of wall-clock training time of backprop over SEDONA as in [1,3]. On ImageNet, we train all models for 600,000 iterations with a batch size of 256 as in the original ResNet paper [4], and below we compare the absolute wall-clock time (hours) for training.
>
> |            | Backprop | DGL  | SEDONA |
> |------------|:--------:|:----:|:------:|
> | ResNet-101 |    113.5 | 59.2 |   56.5 |
> | ResNet-152 | 158.5    | 71.2 | 78.3   |
>
>
> **3. (W3) Bilevel optimization seems to be complicated and would require extensive tuning.**
>
> **A.** Bilevel optimization is only required to run once at the search stage. Once the best decoupled network is found, we can train the network using a greedy block-wise learning, which is faster than normal backprop training with 2x speedup. As in Appendix D.1, we used the same hyperparameters for all architectures at the search stage. This implies that SEDONA can find the best decoupling configuration of various off-the-shelf CNNs with no additional hyperparameter tuning. Moreover, the decoupling configuration is transferable to multiple datasets and additional search is not required for different numbers of blocks $K$.
>
>
>
> **4. (Q2) Given that the gradients used to train each block are a combination of local and backpropped gradients, how does this method avoid the locking problem, and thus provide speedup?**
>
> **A.** Our method uses a combination of local and backpropagated gradients only at the search stage. During training, the network is split into $K$ blocks where each block is trained only with local error signals (i.e. greedy block-wise learning), and we avoid the update locking, meaning that each layer must wait for the signal to propagate through the full network before updating. Section 3.4 describes how the network is split into any given number of blocks, $K$.

---

### Author Response · Authors · 2020-11-23
**The revised draft is now available**

Thank you for your thoughtful feedback. We have tried our best to address as many of comments as possible. We have updated our submission with the following major changes:

1. We added new experimental results on ImageNet with $K=2$ that demonstrate SEDONA can clearly outperform backprop without ensembling (Table 3). In addition, we compared SEDONA to DGL with the setting of the DGL paper when $K=2$ (Appendix C.2).

2. We provided discussions on the effect of discretization and regularization by greedy learning (Appendix C.4-5).

3. We added results of an additional ablation study for comparing SEDONA to DARTS (Appendix C.3).

4. We added detailed descriptions for speedup and ensembling (Section 4.3).

5.  We made a number of other revisions throughout the paper to address other comments such as typos, further clarifications, etc.

We would be happy to hear any additional feedback.

---

### Decision · Program_Chairs · 2021-01-07
**Final Decision**

**Decision:**

Accept (Poster)

**Comment:**

The paper proposes a novel method for greed layer-wise training by considering the learning signal from either backprop or from the additional auxiliary losses. SEarching for DecOupled Neural Architecture learns to identify the decoupled blocks by learning the gating parameters similar to gradient-based architecture search algorithms, such as DARTs.  The empirical experiments demonstrated the effectiveness of SEDONA on CIFAR and TinyImageNet using various ResNet architectures. Several issues of clarity and the correctness of the main theoretical result were addressed during the rebuttal period in the way that satisfied the reviewers. The ideas in this paper are interesting and are broadly applicable. Additional experiments / discussions on the tradeoff between initial search cost and accuracy should be included in the final version.